# The microRNAs Regulating Vascular Smooth Muscle Cell Proliferation: A Minireview

**DOI:** 10.3390/ijms20020324

**Published:** 2019-01-14

**Authors:** Dongdong Wang, Atanas G. Atanasov

**Affiliations:** 1Department of Molecular Biology, Institute of Genetics and Animal Breeding of the Polish Academy of Sciences, 05-552 Jastrzębiec, Poland; 2Department of Pharmacognosy, University of Vienna, Vienna 1090, Austria; 3Institute of Clinical Chemistry, University Hospital Zurich, 8952 Schlieren, Switzerland; 4GLOBE Program Association (GLOBE-PA), Grandville, MI 49418, USA

**Keywords:** miRNA, VSMC, proliferation, atherosclerosis, CVD, restenosis

## Abstract

Vascular smooth muscle cell (VSMC) proliferation plays a critical role in atherosclerosis. At the beginning of the pathologic process of atherosclerosis, irregular VSMC proliferation promotes plaque formation, but in advanced plaques VSMCs are beneficial, promoting the stability and preventing rupture of the fibrous cap. Recent studies have demonstrated that microRNAs (miRNAs) expressed in the vascular system are involved in the control of VSMC proliferation. This review summarizes recent findings on the miRNAs in the regulation of VSMC proliferation, including miRNAs that exhibit the inhibition or promotion of VSMC proliferation, and their targets mediating the regulation of VSMC proliferation. Up to now, most of the studies were performed only in cultured VSMC. While the modulation of miRNAs is emerging as a promising strategy for the regulation of VSMC proliferation, most of the effects of miRNAs and their targets in vivo require further investigation.

## 1. Introduction

Atherosclerosis is the main cause of cardiovascular diseases (CVDs), which are a leading cause of death worldwide [1,2]. Atherosclerosis is an inflammation-associated condition, in which arteries become narrowed and hardened due to an excessive buildup of plaque in the inner lining of the arteries [3,4]. A plaque is made up of cholesterol, other lipids, and diverse cell types [1,2]. At the beginning of the pathologic process of atherosclerosis, vascular smooth muscle cells (VSMCs) migrate from the media into the intima and proliferate within the lesions in response to mediators secreted by monocytes and lymphocytes [5,6]. The monocytes locally differentiate into macrophages, which scavenge the oxidized low-density lipoprotein (oxLDL) to become lipid-laden macrophage foam cells [7,8]. Over years, foam cells and extracellular lipid droplets form a core region of the plaque in the wall of the affected arteries [7,8]. The plaque is covered by a fibrous cap composed of VSMCs and a VSMC-derived extracellular matrix [1,2]. Inflammatory molecules and proteolytic enzymes can weaken the cap, transforming the stable plaque into a vulnerable structure that can rupture [8]. Plaque rupture induces thrombosis and can lead to life-threatening events such as heart attacks and strokes [9]. Therefore, at the beginning of the pathologic process of atherosclerosis, irregular VSMC proliferation promotes plaque formation, but in advanced plaques VSMCs are beneficial, promoting the stability and preventing rupture of the fibrous cap [10].

Aberrant VSMC proliferation represents a significant problem not only during initial plaque formation but also after surgical interventions like percutaneous transluminal coronary angioplasty or bypass surgery contributing to pathological re-narrowing (restenosis) of the affected blood vessel [11]. Restenosis occurring in the first six months after angioplasty has been reported in as many as 25–50% of the patients [12]. Currently, therapy aimed at the inhibition of VSMC proliferation is an established approach against restenosis. The most prominent drugs used in drug-eluting stents so far have been paclitaxel (a microtubule stabilizing agent) and sirolimus (an mTOR inhibitor) [13,14].

Recent studies have demonstrated that microRNAs (miRNAs) are expressed in the vascular system and are involved in the control of VSMC proliferation [15,16]. Each miRNA is able to regulate the expression of multiple target genes, frequently involved in the same cellular pathway. Conversely, target genes may be affected by more than one miRNA. It is presumed that thousands of human genes are targeted by miRNAs [16,17]. Mature miRNAs are short, noncoding ribonucleic acid molecules, typically 22 nucleotides long, which bind to complementary sequences in the 3’-untranslated regions (3’-UTR) of target mRNA transcripts. miRNA biosynthesis is initiated by the transcription of a long capped and poly-adenylated primary miRNA (pri-miRNA) transcript [18,19]. The RNase III enzyme Drosha cleaves the pri-miRNAs into the precursor-miRNAs (pre-miRNAs) [19]. Pre-miRNAs are exported from the nucleus to the cytoplasm, where they are processed by an RNase II enzyme, Dicer. This cleavage event gives rise to a double-stranded product composed of the mature miRNA guide strand and the miRNA passenger strand [20]. Mature miRNAs are then bound to the miRNA-induced silencing complex, which contains two key proteins, argonaute 2 and transactivation-responsive RNA binding protein [16,21]. The mature miRNA and miRNA-induced silencing complex binds to complementary sites in the targeted mRNA transcripts and negatively regulate gene expression. miRNA binding leads to mRNA degradation, inhibition of translation, or both [22,23].

Some miRNAs have been implicated in regulating important VSMC nodal regulators such as cytokines/growth factors and their receptors (e.g., platelet-derived growth factor (PDGF), insulin-like growth factor-1 (IGF-1)), regulators of cell cycle progression (e.g., cyclins, cyclin-dependent kinase inhibitors (CKI), a disintegrin and metalloproteinase with thrombospondin motifs 1 (ADAMTS1)), signaling cascades (e.g., angiotensin II (Ang II)-mediated VSMC signaling pathways, extracellular signal-regulated kinase (ERK)1/2, p38 mitogen-activated protein kinases (MAPK), TGF-β signaling cascades (Smad), signal transducer and activator of transcription (STAT)), transcription factors (e.g., serum-response factor (SRF), Krüppel-like factor-4 (KLF4)), nuclear receptors (e.g., estrogen receptor α (ERα), neuron-derived orphan receptor 1 (NOR1)), and others (e.g., proviral integration site (Pim-1), mammalian target of rapamycin (mTOR)). This minireview briefly summarizes recent findings on the regulatory effects of miRNAs on VSMC proliferation, including both miRNAs that are associated with inhibition or promotion of VSMC proliferation, see Table 1 and Figure 1.

## 2. miRNAs Which Inhibit VSMC Proliferation

### 2.1. miRNAs Influencing Growth Factors/Cytokines, Growth Factor Receptors, and Other Membrane Receptors in VSMCs

It was shown that the increase of miRNA-9 inhibited the serum-induced proliferation of VSMCs by directly targeting the PDGF receptor (PDGFR) disrupting downstream signaling cascades [25]. Furthermore, a small molecule that increased miRNA-9 expression also inhibited neointima formation following balloon injury in vivo [25]. It was demonstrated that the overexpression of miRNA-34a in serum-starved VSMCs significantly inhibited VSMC proliferation and migration, while knockdown of miRNA-34a dramatically promoted VSMC proliferation by targeting neurogenic locus notch homolog protein-1 (Notch1), a single-pass transmembrane receptor [34]. The miRNA-34c inhibited VSMC proliferation and neointimal hyperplasia by targeting the stem cell factor (SCF), a cytokine that binds to the c-kit receptor [35]. miRNA-141 inhibited oxLDL-induced abnormal VSMC proliferation through targeting pregnancy-associated plasma protein A (PAPP-A), a secreted protease targeting IGF-1 binding proteins [40]. A similar effect was also described for another miRNA—miRNA-490-3p, which inhibited the proliferation of VSMCs induced by oxLDL by also targeting PAPP-A [55]. The overexpression of miRNA-206 inhibited VSMC proliferation by silencing the expression of the gap junction protein connexin 43 (Cx43), a component of gap junctions and intercellular channels, via targeting Cx43 3′-UTR [49]. It was reported that miRNA-379 inhibited cell proliferation by targeting 3’-UTR of the *IGF-1* gene [53]. It was shown that miRNA-503 inhibited PDGF-induced human aortic VSMC proliferation and migration by targeting the insulin receptor (INSR), a transmembrane receptor that is activated by insulin, IGF-1 [56]. In addition to PDGFR and INSR, there are other growth factors and their respective receptors which also influence the VSMC proliferation, including fibroblast growth factor (FGF), and epidermal growth factor (EGF). It is very interesting to further investigate the effect of the miRNAs which influence these growth factors and their receptors on VSMC proliferation.

### 2.2. miRNAs Influencing Regulators of Cell Cycle Progression in VSMCs

It was reported that miRNA-22 inhibited VSMC proliferation and migration through repressing ecotropic virus integration site 1 protein homolog (EVI-1) gene expression [29], which is a regulator of cell cycle progression by stabilizing the FBXO5 (F-Box Protein 5) protein and promoting cyclin-A accumulation during interphase [83]. Furthermore, it inhibited neointima formation in wire-injured femoral arteries [29]. Overexpression of miRNA-24 could attenuate VSMC proliferation and neointimal hyperplasia after vascular injuries in diabetic rats. This result is possibly related to the regulation of the expression of Cyclin D1 and CKI p21 through the wingless-type MMTV (Mouse Mammary Tumor Virus) integration site family member 4 (Wnt4)/disheveled-1 (Dvl-1)/β-catenin signaling pathway [32]. A different mechanism of proliferation suppression was demonstrated for miRNA-24, which inhibited high glucose-induced VSMC proliferation and migration by targeting high mobility group box-1 (HMGB1) [84]. A study indicated that miRNA-195 inhibited oxLDL-induced VSMC proliferation by repressing the expression of the cell division control protein 42 homolog (Cdc42), cyclin D1, and fibroblast growth factor 1 (FGF1) genes [48]. The administration of miRNA-195 has been shown to substantially reduce neointima formation in vivo [48]. The upregulation of miRNA-362-3p was demonstrated to inhibit VSMC proliferation and migration, and impede the G_1_/S cell cycle transition by binding to the 3′-UTR of ADAMTS1 and decreasing the levels of its mRNA and protein expression [51]. Further study revealed that significant downregulation of miRNA-362-3p was observed in 110 atherosclerotic coronary artery disease (CAD) patients and not in the 84 control subjects [51]. In addition, miR-365 was reported to inhibit VSMC proliferation partly via modulating the expression of cyclin D1 [52]. Increased levels of miRNA-424/322 inhibited VSMC proliferation in vitro and in injury-induced remodeling in vivo by directly targeting cyclin D1 and calumenin [54]. There are two restriction points in G1/S and G2/M interphases, which ensure correct cell cycle progression [85]. The cell cycle phases are strictly regulated by many regulatory mechanisms. Key regulatory proteins include the cyclins, cyclin-dependent kinases (CDK), CDK inhibitors (CKI), retinoblastoma protein (RB), and the tumor-suppressor gene product [86]. The degree to which the miRNAs influence these factors on VSMC proliferation remains to be examined.

### 2.3. miRNAs Regulating Signaling Cascades in VSMCs

There are some miRNAs regulating signaling cascades in VSMCs. It was shown that the overexpression of miRNA-15b/16 promoted SMC contractile gene expression while attenuating VSMC proliferation by repressing the potent oncoprotein Yes-associated protein (YAP). Knockdown of endogenous miRNA-15b/16 in VSMCs attenuated VSMC-specific gene expression and promoted VSMC proliferation and migration [27]. A different mechanism of proliferation suppression was demonstrated for miRNA-16, which was observed to be highly expressed in VSMCs and to be involved in the Ang-II-mediated VSMC signaling pathways [28]. Lentiviral vector-mediated miRNA-16 knockdown promoted Ang-II-induced cell proliferation and migration, which was associated with the pathways involving ERK1/2 and p38MAPK [28]. It was reported that the expression of miRNA-126 inhibited VSMC proliferation by targeting the low-density lipoprotein receptor-related protein 6 (LRP6) that is involved in a canonical Wnt pathway. Furthermore, it repressed neointima formation in vitro and in vivo [39]. It was reported that the overexpression of miRNA-145-5p inhibited PDGF-induced VSMC proliferation and migration by directly targeting Smad4 and dysregulating the TGF-β signaling cascades, including Smad2, Smad3, and TGF-β [44]. In addition, miRNA-155 was shown to inhibit VSMC proliferation though directly repressing 20-like kinase 2 (MST2) and thus activating the ERK pathway by promoting an interaction between Raf proto-oncogene serine/threonine-protein kinase (Raf-1) and mitogen-activated protein kinase kinase (MEK) [46]. Another study reported that miRNA-155 modulated the proliferation of VSMC by targeting endothelial nitric oxide synthase (eNOS) [47]. The overexpression of miRNA-214 in serum-starved VSMCs significantly decreased VSMC proliferation and migration, whereas knockdown of miRNA-214 dramatically increased the proliferation and migration [50]. Further study indicated that NCK (non-catalytic region of tyrosine kinase adaptor protein 1) associated protein 1 (NCKAP1), involved in the transduction of signals from Ras to Rac, is the functional target of miRNA-214 in VSMCs [50]. Furthermore, upregulation of miRNA-542-3p in old VSMCs significantly inhibited VSMC proliferation, whereas downregulation of miRNA-542-3p in young VSMCs increased VSMC proliferation by targeting spleen tyrosine kinase (Syk)/STAT3-5 axis [57]. It has been shown that the overexpression of miRNA-612 significantly inhibited PDGF-BB-induced migration and invasion of VSMCs through inducing cell cycle arrest at the G_1_ stage by directly decreasing AKT2 (AKT serine/threonine kinase 2) protein expression [58]. The overexpression of let-7d in VSMCs reduced VSMC proliferation by targeting KRAS, which is a member of the small GTPase superfamily and regulates the pathway involved in proliferation [62]. The transfection of let-7g into VSMCs has also been shown to significantly inhibit VSMC proliferation induced by oxLDL through targeting lectin-like oxidized-low-density lipoprotein receptor-1 (LOX-1) [63]. Another study indicated that let-7g maintained VSMC in the contractile status by reducing the PDGF/mitogen-activated protein kinase kinase kinase 1 (MEKK1)/ERK/KLF4 signaling, which further reduced VSMC atherosclerotic change [87]. The signaling cascades in VSMCs are very complicated, including Ras/MAPK cascades, Src-activated signal transduction, phosphatidylinositol 3-kinase (PI3K)/AKT, phospholipase C-γ (PLC-γ), phosphatase and Janus kinase (Jak)/STAT signaling pathways [88]. The influence of the miRNAs regulating other signaling cascades on VSMC proliferation remains to be further investigated in vitro and in vivo.

### 2.4. miRNAs Regulating Transcription Factors in VSMCs

Some miRNAs have been implicated in the regulation of transcription factors. It was demonstrated that miRNA-15a, which is upregulated in VSMCs treated with a transcription factor KLF4, can strongly inhibit the proliferation of VSMCs [26]. Some studies show that miRNA-22-3p overexpression had anti-proliferative and anti-migratory effects by directly targeting HMGB1, a co-factor for gene transcription and proinflammatory factor [89], in human arterial smooth muscle cells (HASMCs) [30]. Furthermore, miRNA-22-3p expression was downregulated and negatively correlated with HMGB1 expression in arteriosclerosis obliterans tissue specimens [30]. miRNA-23b was also shown to suppress the urokinase-type plasminogen activator, Smad3, and a transcription factor forkhead box O4 (FoxO4) expression in phenotypically modulated VSMCs [31]. The overexpression of miRNA-23b in balloon-injured arteries by Ad-miRNA-23b markedly decreased neointimal hyperplasia [31]. Further study validated the transcription factor FoxO4 as a direct target of miRNA-23b in VSMCs [31]. The overexpression of miRNA-124 significantly attenuated PDGF-BB-induced human aortic VSMC proliferation and phenotypic switch by directly targeting the 3′-UTR of the specificity protein-1 (*Sp-1*) gene and then repression of transcription factor Sp-1 expression [36]. MiRNA-124 was further shown to be dramatically downregulated in the aortic media of clinical specimens of the dissected aorta and correlated with molecular markers of the contractile VSMC phenotype [36]. Another study suggested that miRNA-124 inhibited VSMC proliferation by targeting the S100 calcium-binding protein A4 (S100A4), which prevented protein phosphatase 5 (PP5) activation [37]. Additionally, the expression of miRNA-125b was decreased in the arteries with arteriosclerosis obliterans and PDGF-BB-stimulated VSMCs [38]. miRNA-125b suppressed VSMC proliferation and migration but promoted VSMC apoptosis by directly targeting SRF, a member of the MADS (MCM1, Agamous, Deficiens, and SRF) box superfamily of transcription factors [38]. Furthermore, exogenous miRNA-125b expression inhibited vascular neointimal formation in balloon-injured rat carotid arteries [38]. Some studies indicated that the overexpression of miRNA-145 or miRNA-143 is sufficient to promote differentiation and inhibit proliferation of cultured VSMCs by targeting a network of transcription factors, including a transcription factor KLF4, a transcriptional coactivator myocardin, and a transcription activator ELK-1 (member of ETS oncogene family) [41,42]. Deficiency of miRNA-145/143 promoted the synthetic phenotype of VSMCs [90,91]. Adenoviral-mediated gene transfer of miRNA-145/143, which was downregulated after injury [81], inhibited neointimal lesion formation in injured rat carotid arteries [42]. A recent study suggested that miRNA-145 inhibited VSMC proliferation by targeting a member of the tumor necrosis factor (TNF)-receptor superfamily cluster of differentiation 40 (CD40) [43]. It has been demonstrated that overexpression of miRNA-663 potently inhibited PDGF-induced VSMC proliferation and migration. This most likely occurred by inhibition of the expression of a transcription factor Jun B/myosin light chain 9 [60]. Furthermore, adeno-miRNA-663 markedly suppressed the neointimal lesion formation in mice after vascular injury induced by carotid artery ligation, specifically via decreased Jun B expression [60]. Transcription factors are proteins that control the rate of transcription of genetic information from DNA to messenger RNA, by binding to a specific DNA sequence [92]. The miRNA-22-3p regulating HMGB1 might be a very promising target to treat diseases related to VSMC proliferation since its inhibitory effect on VSMCs has been verified in both animal models and humans.

### 2.5. miRNAs Regulating Nuclear Receptors in VSMCs

Some miRNAs have been implicated in regulating nuclear receptors in VSMCs. It was reported that the overexpression of miRNA-152 decreased cell proliferation in LPS-treated VSMCs by downregulating DNA methyltransferase 1 (DNMT1) and subsequently decreasing the methylation of a nuclear receptor ERα gene promoter region [45]. Additionally, miRNA-638 was shown to inhibit PDGF-BB-induced VSMC proliferation and migration by targeting a nuclear receptor NOR1 [59], which is a critical regulator implicated in proliferative vascular diseases [93]. The effect of miRNA-152 on VSMC proliferation in vivo still remains to be further studied.

### 2.6. Others

Some observations suggest that the induction of miRNA-1 by myocardin led to an inhibition of VSMC proliferation by downregulation of Pim-1, which is also named as serine/threonine-protein kinase Pim-1 [20,24]. It was shown that miRNA-761 inhibited Ang-II-induced VSMC proliferation and migration by targeting mTOR, which is a serine/threonine-specific protein kinase [61]. miRNA-29c activation in diabetes mellitus arterial tissues is necessary and sufficient to prevent the exaggerated VSMC growth upon injury [33]. MiRNA-29c overexpression in the injured artery robustly reduced arterial stenosis in diabetes mellitus rats [33]. The targets of miRNA-29c in VSMCs remain to be further investigated.

## 3. miRNAs Which Promote VSMC Proliferation

### 3.1. miRNAs Influencing Growth Factors/Cytokines, Growth Factor Receptors, and Other Membrane Receptors in VSMCs

It has been demonstrated that miRNA-26a promoted VSMC proliferation by directly targeting Smad1 and Smad4, which are two TGF-β- and BMP-related pro-differentiation factors [66]. VSMCdeficient in miRNA-26a shows a significant reduction in proliferation [66]. Some researchers also found miRNA-146b-5p was upregulated in PDGF-BB treated VSMCs. Inhibition of miRNA-146b-5p reduced VSMC proliferation and migration by blocking the VSMC response to PDGF [73]. The detailed molecular mechanisms by which the miRNA-146b-5p regulates the PDGF signaling pathway is so far not clear.

### 3.2. miRNAs Influencing Regulators of Cell Cycle Progression in VSMCs

It was reported that miRNA-17 stimulated the proliferation of VSMCs, enhanced cell cycle G_1_/S transition, and increased levels of proliferating cell nuclear antigen and E2F1 by directly targeting the retinoblastoma (RB) protein mRNA-3′-UTR and then suppressed the expression of RB [64]. In addition, activation of NF-κB p65 resulted in increased miRNA-17 expression in VSMCs, whereas inactivation of NF-κB p65 resulted in decreased expression of miRNA-17 in VSMCs [64]. It was shown that miRNA-25 promoted VSMC proliferation by directly targeting cyclin-dependent kinase 6 (CDK6) [65]. Transfection of VSMC with miRNA-130 decreased the expression of CDKN1A (cyclin-dependent kinase inhibitor 1A) and, in turn, significantly increased smooth muscle proliferation. Conversely, inhibition of miRNA-130 by anti-miRNAs and seed blockers increased the expression of CDKN1A and inhibited VSMC proliferation [70]. The hypoxia-induced miRNA-130a controlled pulmonary SMC proliferation by directly targeting the tumor suppressor p21 (CDKN1A) [70], and growth-arrest-specific homeobox (GAX) [94]. One study indicated that miRNA-208 promoted insulin-induced VSMC proliferation through the downregulation of its potential target p21. However, a miRNA-208 inhibitor alone had no effect on VSMC proliferation [76]. Expression of miRNA-221 and miRNA-222 can be transcriptionally induced by PDGF [77]. The proliferative effect of miRNA-221 and miRNA-222 on VSMC was mediated through silencing of their target proteins, CKI p27Kip1, p57Kip2, and c-kit [77]. Reduction of miRNA-221 blocked the effects of PDGF on the proliferation of VSMCs. In the same line, knockdown of miRNA-221 or miRNA-222 inhibited VSMC proliferation and neointimal formation in rat carotid artery after injury [77]. MiRNA-222 also could promote pulmonary arterial smooth muscle cells (PASMC) proliferation at least partially through targeting p27Kip1 and the tissue inhibitor of metalloproteinase 3 (TIMP3) [78]. Furthermore, miRNA-221 sponge therapy significantly reduced miRNA-221 activity and inhibited neointimal hyperplasia in vein grafts, possibly by targeting p27Kip1 [95]. It is shown that overexpression of miRNA-675 promoted VSMC proliferation in vitro by targeting phosphatase and the tensin homolog (PTEN), which is involved in the regulation of the cell cycle and aggravates restenosis in vivo [80]. So far, only miRNA-221 has been verified to exhibit proliferation-regulating effects on VSMCs in vivo. The effects of other promising miRNAs on VSMC proliferation in vivo need further investigation.

### 3.3. miRNAs Regulating Signaling Cascades in VSMCs

It has been shown that overexpression of miRNA-133 inhibited VSMC proliferation in vitro through an ERK1/2 kinase-dependent pathway and increased expression of Sp-1 [96]. Furthermore, adenoviral overexpression of miRNA-133 in the balloon-injured rat carotid significantly reduced neointimal formation [96]. Key signaling cascades involved in VSMC proliferation include Ras/MAPK cascades, Src-activated signal transduction, PI3K/Akt, PLC-γ, phosphatase and Jak/STAT signaling pathways [88]. The influence of the miRNAs regulating these signaling cascades on VSMC proliferation remains to be further studied in vitro and in vivo.

### 3.4. miRNAs Regulating Transcription Factors in VSMCs

A study suggested that the increased amount of miRNA-29a enhanced VSMC proliferation and promoted atherogenesis, probably through downregulation of Fbw7 (F-box and WD repeat domain-containing 7)/CDC4 (cell division control protein 4) expression by targeting the 3’-UTR of Fbw7/CDC4, subsequently increasing a transcription factor KLF5 stability by reducing the Fbw7/CDC4-dependent ubiquitination of KLF5 [67]. In addition, overexpression of miRNA-146a increased VSMC proliferation by directly targeting KLF4 which could upregulate p21 [72]. Knockdown of miRNA-146a attenuated PDGF-mediated increase of VSMC proliferation [97]. Treatment of balloon-injured rat carotid arteries with antisense oligonucleotides against miRNA-146a resulted in reduced neointima formation and VSMC proliferation in vivo [97]. The reduction in miRNA-200c levels led to increased target gene expression (e.g., *Ubc9* and *KLF4*), which further repressed miRNA-200c levels and accelerated VSMC proliferation [75]. All these miRNAs (miRNA-29a, miRNA-146a, miRNA-200c) targeted a common transcription factor, KLF4, which indicates that KLF4 might be a key target in miRNA-mediated regulation of VSMC proliferation.

### 3.5. Others

In VSMCs cultured in the presence of the proliferation stimulator PDGF-BB, miRNA-29b was upregulated significantly [68]. miRNA-29b is demonstrated as being an important regulator in the PDGF-BB-mediated VSMC phenotypic transition by targeting SIRT1 (gene encoded Sirtuin-1, also known as NAD-dependent deacetylase sirtuin-1) [68]. The same target was also described for another miRNA. An in vivo study shows that miRNA-138 promoted VSMC proliferation and migration in db/db mice through downregulation of SIRT1 [71]. Additionally, the inhibition of miRNA-204 in diabetes mellitus arterial tissues has been shown to prevent the exaggerated VSMC growth upon injury [33]. Moreover, miRNA-204 inhibition in the injured artery significantly reduced arterial stenosis in diabetes mellitus rats [33]. Other studies indicated that miRNA-204 targets Sirtuin-1 in endothelial cells [98,99]. However, the targets of miRNA-204 in VSMCs remain to be further studied. miRNA-31 has been proven to promote the VSMC contractile phenotype by repressing the cellular repressor of E1A-stimulated genes (CREG) expression, a secreted glycoprotein that inhibits cell growth [69]. It was reported that miRNA-574-5p expression was elevated in the sera and VSMCs of patients with CAD [79]. Additionally, miRNA-574-5p overexpression promoted cell proliferation and inhibited apoptosis in VSMCs by directly targeting *ZDHHC14* (Zinc Finger DHHC-Type Containing 14) gene [79]. Overexpression or transfection of miRNA-155-5p mimic elevated the proliferation and migration of VSMCs, which was blocked by treatment with an inhibitor of miRNA-155-5p [74]. The targets of miRNA-155-5p in VSMCs require further investigation. Both miRNA-29b and miRNA-138 targeted SIRT1 to regulate VSMC proliferation, which suggests that SIRT1 might be a key target mediating the miRNA-modulatory effects on VSMC proliferation.

## 4. miRNA-21 Which Promotes and Inhibits VSMC Proliferation

Some studies suggest that the function of miRNA-21 is likely to be complex and highly context-dependent. In addition to promoting contractile gene expression, miRNA-21 has been found to promote VSMC proliferation [81]. MiRNA-21 is demonstrated to promote VSMC proliferation by silencing PTEN, a tumor suppressor protein, and increasing B-cell lymphoma 2 (Bcl-2), which increased VSMC proliferation and survival [72,81]. In the same line, a depletion of miRNA-21 caused decreased VSMC proliferation. Local delivery of an antisense oligonucleotide to knockdown miRNA-21 inhibited neointima formation in a rat carotid artery after angioplasty [72,81]. The decrease of the expression of miRNA-21 by adenovirus-mediated miRNA-21 sponge gene therapy significantly reduced the proliferation in cultured VSMCs and the proliferation rates in grafts compared with controls at 28 days after bypass surgery [100]. miRNA-21 sponge gene transfer therapy also reduced the intimal/media area ratio in vein grafts compared with the controls and improved vein graft hemodynamics probably by targeting PTEN in vein grafts [100]. On the other hand, in some studies, the increased miRNA-21 is shown to promote VSMC differentiation and inhibit VSMC proliferation by upregulating VSMC-restricted contractile proteins by silencing programmed cell death protein 4 (PDCD4), a tumor suppressor protein [82]. To understand the complicated role of miRNA-21 in the regulation of VSMC proliferation, further studies need to be performed.

## 5. Conclusions

Extensive research during the last decade investigated the association of miRNAs with VSMC proliferation. In this minireview, we briefly summarize the miRNAs that regulate VSMC proliferation. These miRNAs exert their regulation of VSMC proliferation by regulating a number of target proteins and signaling cascades. The targets modulated by miRNA implied in the inhibition of VSMC proliferation include PDGFR, KLF4, YAP, EVI-1, HMGB1, Cyclin D1, p21, Sp-1, S100A4, SRF, LRP6, PAPP-A, myocardin, ELK-1, *ERα* gene, Cx43, NCKAP1, ADAMTS1, cyclin D1, IGF-1, AKT2, *Notch1* gene, PDGFR downstream signaling cascades, Ang-II-mediated VSMC signaling pathways, TGF-β signaling cascade (Smad2, Smad3, Smad4, and TGF-β), ERK1/2, p38MAPK, Wnt4/Dvl-1/β-catenin signaling pathway, Syk/ STAT3-5 axis, DNMT1, MST2, Raf-1, MEK, eNOS, Cdc42, cyclin D1, FGF1, INSR, NOR1, Jun B/myosin light chain 9, mTOR, KRAS, LOX-1 and MEKK1/ERK/KLF4 signaling. The targets modulated by miRNAs implied in increased proliferation of VSMC include the RB protein, NF-κB p65, CDK6, TGF-β- and BMP-related pro-differentiation factors (Smad1 and Smad4), Fbw7/CDC4, KLF5, SIRT1, CREG, CDKN1A, GAX, Sp-1, ERK1/2 kinase-dependent pathway, SIRT1, KLF4, p21, Ubc9, CKI p27Kip1, p57Kip2, and c-kit, ZDHHC14, PTEN, Bcl-2, and PDCD4. The majority of the studies up to now were performed only in cultured VSMC. The effects of these miRNAs and their targets in vascular muscle tissue in vivo need to be further investigated. Furthermore, there is one open question on the specificity of miRNA action for VSMCs. To this end, there are some tools (such as TissueAtlas [101] and IMOTA (https://ccb-web.cs.uni-saarland.de/imota/)) that have been generated and can be used to check whether the miRNAs and their target genes are expressed in specific tissues [102]. Additionally, endothelial barriers play a very important role in atherosclerosis, restenosis, and other CVDs. Targeting of VSMC proliferation may influence the endothelial cell layer by modulating the proliferation of endothelial cells. Therefore, when investigating the effects of miRNAs on VSMC proliferation, it is also of relevance to examine their effects on endothelial cell proliferation.

## Figures and Tables

**Figure 1 ijms-20-00324-f001:**
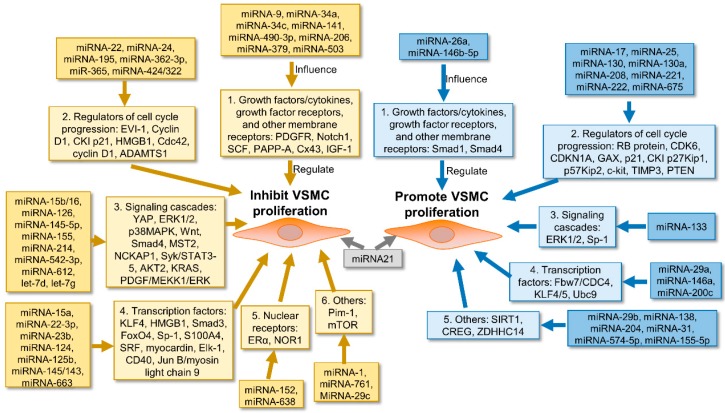
The role of miRNAs in the regulation of vascular smooth muscle cell (VSMC) proliferation.

**Table 1 ijms-20-00324-t001:** miRNAs implied in the regulation of VSMC proliferation.

miRNA	Implied Targets	References
***miRNAs which inhibit VSMC proliferation***		
miRNA-1	Proviral integration site (Pim-1)	[20,24]
miRNA-9	Platelet-derived growth factor receptor (PDGFR) and further downstream signaling cascades	[25]
miRNA-15a	Krüppel-like factor-4 (KLF4)	[26]
miRNA-15b/16	The potent oncoprotein yes-associated protein (YAP) and the pathways extracellular signal-regulated kinase (ERK)1/2 and p38MAPK (mitogen-activated protein kinases)	[27,28]
miRNA-22	Ecotropic virus integration site 1 protein homolog (EVI-1)	[29]
miRNA-22-3p	High mobility group box-1 (HMGB1)	[30]
miRNA-23b	The transcription factor forkhead box O4 (FoxO4)	[31]
miRNA-24	The wingless-type Mouse Mammary Tumor Virus (MMTV) integration site family member 4 (Wnt4)/disheveled-1 (Dvl-1)/β-catenin signaling pathway	[32]
miRNA-29c	N.A.	[33]
miRNA-34a	Neurogenic locus notch homolog protein-1 (Notch1)	[34]
miRNA-34c	Stem cell factor (SCF)	[35]
miRNA-124	The 3′-UTR of the specificity protein-1 (Sp-1) gene or S100 calcium-binding protein A4 (S100A4)	[36,37]
miR-125b	Serum-response factor (SRF)	[38]
miRNA-126	Low-density lipoprotein receptor-related protein 6 (LRP6)	[39]
miR-141	Pregnancy-associated plasma protein A (PAPP-A)	[40]
miRNA-143/145	KLF4, myocardin, ELK-1, and cluster of differentiation 40 (CD40)	[41,42,43]
miRNA-145-5p	Smad4 and the TGF-β signaling cascades, including Smad2, Smad3 and TGF-β	[44]
miRNA-152	DNA methyltransferase 1 (DNMT1) and the methylation of ERα gene promoter region	[45]
miR-155	20-like kinase 2 (MST2), the ERK pathway, or endothelial nitric oxide synthase (eNOS)	[46,47]
miRNA-195	The Cdc42, cyclin D1, and fibroblast growth factor 1 (FGF1) genes	[48]
miRNA-206	3′-UTR of the gap junction protein connexin 43 (Cx43)	[49]
miRNA-214	NCK associated protein 1 (NCKAP1)	[50]
miRNA-362-3p	A disintegrin and metalloproteinase with thrombospondin motifs 1 (ADAMTS1)	[51]
miRNA-365	Cyclin D1	[52]
miR-379	3′-UTR of insulin-like growth factor-1 (IGF-1)	[53]
miRNA-442/322	Cyclin D1 and calumenin	[54]
miRNA-490-3p	PAPP-A	[55]
miRNA-503	Insulin receptor (INSR).	[56]
miRNA-542-3p	Spleen tyrosine kinase (Syk)/signal transducer and activator of transcription (STAT)3-5 axis	[57]
miRNA-612	AKT2 protein	[58]
miRNA-638	Neuron-derived orphan receptor 1 (NOR1)	[59]
miRNA-663	JunB/myosin light chain 9	[60]
miRNA-761	Mammalian target of rapamycin (mTOR)	[61]
let-7d	KRAS (Kirsten rat sarcoma 2 viral oncogene homolog)	[62]
let-7g	Lectin-like oxidized-low-density lipoprotein receptor-1 (LOX-1), and PDGF/mitogen-activated protein kinase kinase kinase 1 (MEKK1)/ ERK/KLF4 signaling	[63]
***miRNAs which promote VSMC proliferation***		
miRNA-17	Retinoblastoma (RB) protein mRNA-3′-UTR	[64]
miRNA-25	Cyclin-dependent kinase 6 (CDK6)	[65]
miRNA-26a	Smad1 and Smad4	[66]
miRNA-29a	3’-UTR of Fbw7/CDC4, KLF	[67]
miRNA-29b	SIRT1	[68]
miRNA-31	Cellular repressor of E1A-stimulated genes (CREG) expression.	[69]
miRNA-130	The tumor suppressor p21 (CDKN1A)	[70]
miRNA-130a	CDKN1A, and growth arrest-specific homeobox (GAX).	[70]
miRNA-133	CDKN1A	[70]
miRNA-138	SIRT1	[71]
miRNA-146a	KLF4	[72]
miRNA-146b-5p	The response to PDGF	[73]
miRNA-155-5p	N.A.	[74]
miRNA-200c	Ubc9 and KLF4	[75]
miRNA-204	N.A.	[33]
miRNA-208	p21	[76]
miRNA-221	PDGFR	[77]
miRNA-222	p27Kip1 and tissue inhibitor of metalloproteinase 3 (TIMP3)	[78]
miRNA-574-5p	*ZDHHC14* (Zinc Finger DHHC-Type Containing 14) gene	[79]
miRNA-675	PTEN	[80]
***miRNA which promotes and inhibits VSMC proliferation***		
miRNA-21	Promotion of VSMC proliferation: phosphatase and tensin homolog (PTEN), B-cell lymphoma 2 (Bcl-2).	[72,81]
	Inhibition of VSMC proliferation: Programmed cell death 4, a tumor suppressor protein.	[82]

N.A. = not available.

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
