# Peer review of "The microRNAs Regulating Vascular Smooth Muscle Cell Proliferation: A Minireview"

_ijms, 2019, doi:10.3390/ijms20020324_

Reviewer 1 Report

Wang and Atanasov provide a very comprehensive on microRNAs that regulate vascular smooth muscle cell proliferation. The information on the core topic is already very comprehensive and “minireview” almost a understatement. 

While for the main topic I did not found any manuscript in the literature missing (I only did however a quick check) I have one suggestion for the authors. miRNA targeting seems to be much more complex as assumed. Recently it has been shown that there is in some cases even a deterministic role of 5-mers. Moreover, miRNAs seem to jointly work in pathways, as the authors also nicely indicate. There exist complete dictionary on microRNAs and target pathways. These points together open the question on the specificity of miRNA target gene regulation. This fact could be added in a short paragraph. There are from my perspective two points to be taken into account of which the second one is the more important one:  a) whether the proposed regulations are indeed happening in vascular muscle and b) whether they are specific for vascular cells. To this end, tools as the miRNA tissue atlas have been generated to check whether the miRNAs are expressed or even specific for heart muscle. Even further, tools as IMOTA allow to check whether the miRNA and the target gene are expressed in a certain tissue. There exist many miRNA target gene pairs in the literature claimed for one tissue and neither the miRNA nor the gene are expressed in that tissue.   

I would leave it to the authors to add a paragraph on these challenges, the review is already very comprehensive. My intention was only to mention one of the current challenges in miRNA-targeting. The more people do background checks the less false positive miRNA gene interactions we will hopefully see in the future. 

Author Response

We thank the reviewer for the time taken to evaluate our work and for the very nice suggestions! We have added the challenges about specificity of miRNA target genes in the manuscript (Page 9).

Reviewer 2 Report

In this review article, the authors reviewed the literature related to the role of microRNAs in vascular smooth muscle cell proliferation. The authors detailed the various miRNAs involved in multiple processes associated with vascular smooth muscle cell proliferation. This is an exciting and timely review article. The authors have covered various aspects of the topic. The manuscript is very well written and presented in a logical manner. Most of the discussed topics in the review describe what is known, but critical inputs are not provided. The author may discuss the known fact for each topic, then they should provide their own perspectives on each topic, how future research may further advance this area of research.

Author Response

We thank the reviewer for the time taken to evaluate our work and for the very nice suggestions! We have added our own perspectives on each topic and now give our view about how future research may further advance this area (Pages 3-9).

Reviewer 3 Report

In this manuscript, the authors summarized the role of miRNAs regulating the proliferation of Vascular smooth muscle cell (VSMC) which is critical for the development of atherosclerosis. Authors described the recent findings demonstrating the role of miRNAs promoting or inhibiting proliferation of VSMC as well as miRNAs which can promote and inhibit VSMC proliferation. Authors also acknowledge that most of the studies on miRNAs were performed in the cultured VSMC system and their role in the in vivo system remains to be explored.

The article appears to be carefully prepared and the text is easy to follow. The table is very informative. However, I have a few comments to improve the manuscript.

Comments:

1.     A figure illustrating the role of the miRNAs in VSMC proliferation is necessary.

2.     Adding another Figure showing the role of VSMC proliferation in atherosclerosis will be helpful. 

Author Response

We thank the reviewer for the time taken to evaluate our work and for the very nice suggestions!

1. We have added a figure to illustrate the role of miRNAs in the regulation of VSMC proliferation (Figure 1).

2. Since there are many reviews or papers including figures to show the role of VSMC proliferation in atherosclerosis, we cited these reviews to avoid repeating information in this minireview (Page 1).